# Injectable Hydrogel Based on Protein-Polyester Microporous Network as an Implantable Niche for Active Cell Recruitment

**DOI:** 10.3390/pharmaceutics14040709

**Published:** 2022-03-26

**Authors:** V.H. Giang Phan, Mohanapriya Murugesan, Panchanathan Manivasagan, Thanh Loc Nguyen, Thuy-Hien Phan, Cuong Hung Luu, Duy-Khiet Ho, Yi Li, Jaeyun Kim, Doo Sung Lee, Thavasyappan Thambi

**Affiliations:** 1Biomaterials and Nanotechnology Research Group, Faculty of Applied Sciences, Ton Duc Thang University, Ho Chi Minh City 70000, Vietnam; phanvuhoanggiang@tdtu.edu.vn (V.H.G.P.); thuyhien.phan.w@gmail.com (T.-H.P.); luuhungcuong145@gmail.com (C.H.L.); 2Graduate School of Biotechnology, College of Life Sciences, Kyung Hee University, Yongin-si 17104, Gyeonggi-do, Korea; priyabioinfo@khu.ac.kr; 3Department of Applied Chemistry, Kumoh National Institute of Technology, Daehak-ro 61, Gumi-si 39177, Gyeongbuk, Korea; manimaribtech@kumoh.ac.kr; 4School of Chemical Engineering, Sungkyunkwan University, Suwon 16419, Gyeonggi-do, Korea; nguyenloc2805@gmail.com (T.L.N.); kimjaeyun@skku.edu (J.K.); 5Department of Bioengineering, University of Washington, Seattle, WA 98195, USA; duykhiet.ho@gmail.com; 6College of Materials and Textile Engineering, Nanotechnology Research Institute, Jiaxing Unviersity, Jiaxing 314001, China; liyi@zjxu.edu.cn

**Keywords:** thermo-responsive copolymers, sol-gel phase transition, injectable hydrogels, protein-polymer conjugation, immune cell recruitment, dendritic cells

## Abstract

Despite the potential of hydrogel-based localized cancer therapies, their efficacy can be limited by cancer recurrence. Therefore, it is of great significance to develop a hydrogel system that can provoke robust and durable immune response in the human body. This study has developed an injectable protein-polymer-based porous hydrogel network composed of lysozyme and poly(ε-caprolactone-co-lactide)-*b*-poly(ethylene glycol)-*b*-poly(ε-caprolactone-co-lactide (PCLA) (Lys-PCLA) bioconjugate for the active recruitment dendritic cells (DCs). The Lys-PCLA bioconjugates are prepared using thiol-ene reaction between thiolated lysozyme (Lys-SH) and acrylated PCLA (PCLA-Ac). The free-flowing Lys-PCLA bioconjugate sols at low temperature transformed to immovable gel at the physiological condition and exhibited stability upon dilution with buffers. According to the in vitro toxicity test, the Lys-PCLA bioconjugate and PCLA copolymer were non-toxic to RAW 263.7 cells at higher concentrations (1000 µg/mL). In addition, subcutaneous administration of Lys-PCLA bioconjugate sols formed stable hydrogel depot instantly, which suggested the in situ gel forming ability of the bioconjugate. Moreover, the Lys-PCLA bioconjugate hydrogel depot formed at the interface between subcutaneous tissue and dermis layers allowed the active migration and recruitment of DCs. As suggested by these results, the in-situ forming injectable Lys-PCLA bioconjugate hydrogel depot may serve as an implantable immune niche for the recruitment and modification of DCs.

## 1. Introduction

Hydrogels, as water-swollen three-dimensional amphiphilic or hydrophilic polymer networks, are used for the encapsulation and controlled release of chemotherapeutic drugs, therapeutic proteins as well as peptides, nucleic acid and cells [1,2,3,4]. Owing to these features, hydrogels are actively exploited in immunotherapy and regenerative medicine [5,6]. In fact, the hydrophilic and porous nature of hydrogels has provided oxygen, metabolites and nutrients permeability to network [7,8,9]. Their high swelling nature provides flexibility to the hydrogels, yet they are water-insoluble at the implantation site [10]. Numerous techniques have been explored to prepare hydrogels with distinct characteristics including pore size, mechanical strength, and biodegradation [11,12,13]. In addition, the choice of polymer and the density of physical and chemical cross-linking certainly control the biocompatibility, biodegradation and more importantly the release kinetics of the hydrogel network [14,15]. In addition, synthetic polymers provide structural flexibility and enhanced mechanical properties to the hydrogel network [16,17]. However, the improved physical properties are limited by poor cell adhesion, biocompatibility, and hydrolytic and enzymatic degradability [18]. Interestingly, the natural polymers are the choice of interest with minimal inflammatory response and ability to stimulate specific immune response [19]. Hydrogels prepared using natural polymers such as chitosan, zein and gelatin by cross-linking exhibited good stimuli-responsive characteristics and controlled release characteristics [20,21,22,23]. In addition, poly(caprolactone)-based synthetic hydrogels showed sharp sol-to-gel phase transition and prolonged the release of therapeutics [24].

It has been found that the implantation of hydrogels into the body is a suitable strategy to cure various kinds of acute and chronic diseases [25], but the implantation of preformed hydrogels to the body requires surgery that is limited by its high cost and poor patient compliance [26]. Therefore, polymers sols that could transform into an in-situ cross-linked hydrogel network upon exposure to the physiological stimuli (e.g., pH and temperature) have received much attention due to their sharp phase transition [27,28]. Unlike the conventional materials that required organic solvents to load therapeutic agents, the in situ forming hydrogel precursors could effectively imbibe drugs, proteins, nucleic acids or cells by simple mixing [29,30,31].

In recent years, injectable hydrogels and scaffolds have been explored in a wide range of immunological applications [32,33,34]. Generally speaking, the in situ implanted hydrogels containing appropriate antigens or cytokines effectively recruit, migrate, and activate immune cells in human body [35,36]. This is unlike ex vivo immune modulation of cancer patients that requires complex designs, in which patients’ own immune cells are isolated and activated ex vivo [37]. Through the use of injectable hydrogels, the free-flowing sols could be injected into the patient of interest and could generate antigen-specific immune response via the introduction of appropriate antigens [38,39]. The release of antigens from the microporous hydrogel network can stimulate antigen-presenting cells, including dendritic cells (DCs), macrophages, and B cells [33,40]. Owing to the unique features of DCs, such as superior ability to take up, process and present antigens on both MHC II and MHC I molecules to CD^4+^ and CD^8+^ cells, respectively, they are considered as the key regulators in initiating the immune response [41]. Hence, hydrogels that enhance antigen uptake and trigger the maturation of DCs for cancer immunotherapy have received much attention.

This study proposed a thermosensitive poly(ε-caprolactone-co-lactide)-*b*-poly(ethylene glycol)-*b*-poly(ε-caprolactone-co-lactide (PCLA-PEG-PCLA) (hereafter referred as PCLA) conjugated lysozyme (Lys-PCLA) bioconjugate for the active recruitment and modulation of dendritic cells (DCs) (Figure 1). Lysozyme (Lys), as a 14.4 kDa molecular weight stable and abundant protein found in the living organisms [42], maintains its three-dimensional structure in a range of pH from five to nine. Compared with its unmodified natural counterpart, the polymer modified Lys showed improved pharmacokinetics [43]. In addition, the presence of abundant amine functional groups on the surface of Lys allows chemical modification or physical assembly with polymers for further processes [44]. Lys exhibits antiviral properties by catalyzing the hydrolysis of β-1,4 glycosidic bonds between *N*-acetylglucosamine and *N*-acetylmuramic acid in the cell wall of bacteria [45]. In this research, the Lys was chemically conjugated with PCLA copolymer by the thiol-ene chemistry. Beyond that, the in situ gelation ability and in vitro toxicity of Lys-PCLA injectable hydrogel were studied by exposing it to RAW 263.7 cells. Sprague-Dawley (SD) rats were used to examine an in vivo in situ gel formation. Furthermore, the active recruitment of immune cells to Lys-PCLA-based bioconjugate network was investigated after subcutaneous implantation of the depot onto the back of mice, and the recovery of the hydrogel was analyzed three days after injection. 

## 2. Materials and Methods

### 2.1. Materials

Lys from chicken egg white, tris(2-carboxyethyl)phosphine hydrochloride salt (TCEP), PEGs (M_n_ = 1500 g/mol), DL-lactide (LA), ε-caprolactone (CL), triethyl amine (TEA), acryloyl chloride, and stannous octoate (Sn(Oct)_2_) were supplied by Sigma-Aldrich (St. Louis, Mo, USA).

### 2.2. Preparation of Lys-PCLA Bioconjugates

The Michael reaction between acrylated PCLA (PCLA-Ac) and thiolated Lys (Lys-SH) was used to prepare Lys-PCLA bioconjugate. Meanwhile, the synthetic route to obtain Lys-PCLA bioconjugate is shown in Figure 1.

*Preparation of PCLA copolymer:* The PCLA copolymer was synthesized by following the previously reported procedure [46]. Briefly speaking, PEG (10 g, 6.1 mmol) and Sn(Oct)_2_ (0.1 g, 0.25 mmol) were placed and stirred for 3 h at 110 °C. Afterwards, the temperature was decreased to 60 °C. Subsequently, CL (19.5 mL, 166 mmol) and LA (7.5 g, 52 mmol) were added and allowed to polymerize at 130 °C for 18 h. One day after the reaction, chloroform was added to reduce the viscosity of the reaction mixture and the product was precipitated using the *n*-hexane and diethyl ether mixture (1:1, *v*/*v*). Yield: 79%.

*Preparation of PCLA-Ac:* Hydroxyl chain-end of the PCLA copolymer reacted with acryloyl to introduce acrylate groups. Briefly speaking, PCLA (1 mmol) was dissolved in toluene in the presence of TEA (3 mmol). Apart from that, the solution was stirred at 0 °C. Acryloyl chloride (3 mmol) in toluene was added dropwise and then stirred for 48 h. Furthermore, the PCLA-Ac copolymer was obtained by precipitating the reaction mixture in diethyl ether. Yield: 89%.

*Preparation of Lys-SH:* Lys was dissolved in 2 M urea containing PBS buffer solution at a concentration 5 mg/mL under the nitrogen environment. Afterwards, TCEP (14 eq.) was added adjusted to 8.0 in terms of pH and stirred at 0 °C for 1 h. The Lys-SH was obtained by passing it through a PD-10 desalting column and then the product was stored at 0 °C. Yield: 92%.

*Preparation of Lys-PCLA bioconjugates:* The Lys-PCLA bioconjugate was prepared using the Michael-addition reaction between Lys-SH (1 mmol) and PCLA-Ac (3 mmol). In brief, PCLA-Ac was dispersed in PBS at 0 °C (10 wt.% concentration). Subsequently, Lys-SH was added and stirred at 0 °C. After two days, the reaction mixture was moved to the cellulose membrane (MWCO: 3500 kDa) and extensively dialyzed against deionized water for three days. In order to avoid the Lys-PCLA bioconjugate aggregation, the dialysis was performed at 0 °C. Finally, Lys-PCLA bioconjugate was acquired by lyophilization. Yield: 76%.

### 2.3. Characterization

*^1^H NMR analysis:* The chemical structure and number-average molecular weight (M_n_) of copolymers were confirmed using ^1^H NMR spectra (Varian Unity Inova 500 SNB, Palo Alto, CA, USA). In order to record ^1^H NMR, the PLCA copolymer or PLCA-Ac copolymer (1 wt.%) had been dissolved in CDCl_3_.

*FT-IR:* FT-IR spectra of the copolymer and bioconjugate were obtained using an FT-IR instrument (Termo Scientifc, Waltham, MA, USA) within the scanning range of 400–4000 cm^−1^.

*Scanning electron microscope (SEM):* The surface morphology of hydrogels was observed using SEM (SEM, JEOL JSM-6390, Austin, TX, USA). After hydrogel samples were placed on SEM stubs, samples were sputter-coated with platinum to make them conductive.

*Gel permeation chromatography (GPC):* The number-average molecular weight (M_n_) and polydispersity index (PDI) of PCLA copolymers were measured using Waters GPC (Milford, MA, USA) using polystyrene standards.

### 2.4. Sol-Gel Phase Transition Behavior

The tube inversion method was adopted to investigate the flow (sol) and non-flow (gel) behavior of PCLA copolymers and Lys-PCLA bioconjugates [47]. In order to obtain a sol-gel phase diagram, different weights of copolymers in PBS (pH 7.4) were placed in a 4 mL vial and stirred at 0 °C for 12 h. After preparation of the homogenous suspension, the vials were placed in a thermo-sensitive water bath and the temperature was increased at a constant rate (2 °C increment with 20 min interval). Apart from that, the flow (sol) and non-flow (gel) nature of the copolymers had been confirmed by inverting the tube. The copolymers were considered as gel when there was no flow after 2 min. The water contact angle was measured using a previously reported procedure [48].

### 2.5. Rheology Measurement

In order to examine the mechanical strength of hydrogels prepared from PCLA copolymers and Lys-PCLA bioconjugates, the mechanical properties including viscosity, storage modulus (G′), and loss modulus (G″) were measured using a Bohlin Rotational Rheometer equipped with a parallel steel plate (40 mm) and a Peltier system to control the temperature. For the rheological measurement, 22.5 wt.% of either PCLA copolymer or Lys-PCLA conjugate was prepared, as described in the previous section, and the samples were placed between the parallel plates with a plate gap of 0.25 mm.

The gelation temperature (T_gel_) was determined by raising the temperature from 10 °C to 60 °C at a constant frequency (1 Hz) and shear stress (0.4 pa) with a heating rate of 2 °C/min. The G′ and G″ were measured as a function of temperature and the T_gel_ was determined from the intersection point of the curves. The complex viscosity was also measured without changing the measurement parameters.

In order to determine the linear viscoelastic region (LVER) and observe the point at which the hydrogel structure begin to deteriorate, the strain sweep experiments were performed from 1% to 100% at a constant frequency (1 Hz) and temperature (37 °C). The frequency sweep test was also conducted from 0 to 50 Hz within the LVER.

### 2.6. In Vitro Cytotoxicity and Imaging Cell Viability

The RAW 263.7 cells were purchased from Korean Cell Line Bank (KCLB) and cultured in a DMEM supplemented with 10% fetal bovine serum and 1% penicillin-streptomycin (1%, (*w*/*v*)). Beyond that, the cell flasks were incubated at 37 °C in a humidified 5% CO_2_-95% air atmosphere.

In order to determine the biocompatibility of copolymers and bioconjugates, the RAW 263.7 cells, at a density of 10^4^ cells, were cultured with various concentrations of copolymers in a 96-well plate. After 2 days, 20 µL of MTT solution (from 5 mg/mL stock solution) was added and incubated for 3 h at 37 °C. At the same time, the purple crystals were dissolved in DMSO and the viability was examined with a Microplate reader by measuring the absorbance at 490 nm. The RAW 263.7 cells cultured with only fresh culture medium were taken as the control.

### 2.7. In Situ Gelation In Vivo

In order to investigate the in situ gelation properties of hydrogels, six-week-old SD rats with an average weight of 225 g were employed. In vivo gelation was confirmed by subcutaneously administering 300 µL (22.5 wt.%) of PCLA copolymers or Lys-PCLA conjugate solutions into the back of SD rats. Shortly before use, the hydrogel precursors were transferred to a hypodermic syringe equipped with a 26G needle and kept in an ice bath to avoid early gelation. In addition, the injection site of the SD rats was shaved and sterilized using 70% ethanol. Ten minutes after the injection, the SD rats were sacrificed. The injection site was cut-open to recover the hydrogels recovered and photographed. Afterwards, the recovered hydrogels were frozen and freeze-dried. The freeze-dried products were cross-sectioned and coated with platinum using a spin coater and then the porous structure was visualized using SEM.

### 2.8. In Vivo Cell Recruitment by Hydrogels

In order to analyze the recruitment of the cell in the hydrogel network, either Lys-PCLA solution or PCLA solution (200 µL, 22.5 wt.%) was subcutaneously injected to BALB/c mice and allowed to recruit cells for 3 days. Thereafter, the hydrogel nodules were collected and the cells were counted by digesting hydrogels with collagenase type II solution. Furthermore, the hydrogels debris was filtered by a 20 µm cell trainer and subsequently washed twice with a buffer. Afterwards, FcR blocking reagent was adopted to count and stain the cells. Subsequently, the cells were stained using mouse monoclonal antibodies against CD11c and CD11b in a FACS buffer followed by flow cytometry analysis.

All the animal experiments were followed according to the Sungkyunkwan University standards protocols and the Institutional Committees of Sungkyunkwan University approved all the animal experiments performed in this study.

### 2.9. Statistical Analysis

The statistical significance between the groups is analyzed using a two-tailed Student *t*-test.

## 3. Results and Discussion

The design of injectable hydrogels that could recruit and modulate immune cell response has been a hot spot of research [49]. Among them, the thermo-responsive copolymers that exhibit sharp phase transition have been a polymer of interest in various biomedical applications [50]. Thermo-responsive polymers, such as pluronic, poly(*N*-isopropylacryamide) and polyphosphazene, have often been studied because of their sharp transition properties [51]. In our group, we extensively studied the polyester-based PCLA copolymer as a thermo-responsive copolymer due to its sharp transition, good biocompatibility and controlled biodegradability without toxicity at the implanted site [52,53,54]. Herein, we conjugated the PCLA copolymers to the Lys to recruit immune cells into the microporous network.

### 3.1. Synthesis and Characterization of Lys-PCLA Bioconjugate

The synthesis route to prepare the Lys-PCLA bioconjugate is shown in Figure 1. Firstly, the PCLA copolymer was prepared by ring-opening polymerization of CL and LA with a PEG macroinitiator. In addition, the PCLA copolymer formation and structure was confirmed using ^1^H NMR spectra. ^1^H NMR spectra show the PCLA copolymer featuring characteristics peaks of PEG, CL and LA components (Figure 2A). The characteristic PEG methylene protons (-O-CH_2_-CH_2_-O-) appeared at 3.63 ppm, whereas the methylene proton next to the carbonyl appeared at 2.29 ppm. At the same time, the methine proton that originated in lactide units appeared at 5.13 ppm. The molar composition and M_n_ of the PCLA copolymer was calculated by comparing the characteristic peaks of PEG at 3.63 ppm with CL and LA characteristic peaks at 2.29 ppm and 5.13 ppm, respectively. Then, PCLA-Ac was prepared by acrylation of chain-end hydroxyl functional groups in the presence of triethylamine base. ^1^H NMR shows the appearance of new peaks at 5.76 to 6.66 ppm corresponding to the characteristic peaks of acrylate, which indicates successful acrylation of PCLA copolymers (Figure 2B). GPC trace in Figure 2C also confirmed the successful PCLA copolymer synthesis. From the ^1^H NMR spectra and GPC trace, it is confirmed that CL/LA ratio and PDI of PCLA copolymer was found to be 2.37 and 1.38, respectively (Table 1). Furthermore, FT-IR spectra of PCLA copolymer confirm the successful synthesis and the presence of characteristic functional groups (Figure 2D). 

In the second step, the Lys was reacted with mild reducing agent TCEP to obtain Lys-SH that allows the effective conjugation with PCLA. In general, the native chicken egg white Lys contains eight cystein residues (i.e., four disulfide bridges) [55]. The presence of disulfide bonds in the Lys plays an important role in maintaining the protein structure and the hydrophobic cavities within the protein [56]. In this study, for effective conjugation of Lys amphiphilic PCLA copolymer, the disulfide bond in the Lys was reduced with TCEP. Apart from that, the reduction of Lys allows the unfolding of the polypeptide chain and increases the water solubility [18]. The reduction of Lys was confirmed using Ellman’s test. The presence of free thiol concentration increased with the increasing concentration of TCEP (data not shown). Then, the conjugation of Lys-PCLA bioconjugate was obtained by reacting Lys-SH and PCLA-Ac via the Michael addition reaction.

### 3.2. Sol-to-Gel Phase Diagram

The classical tube inversion method was used to determine the phase diagram of PCLA copolymer and Lys-PCLA bioconjugate in aqueous solutions [47]. As shown in Figure 3A, both polymers exhibit sol-to-gel phase transition with increasing temperature and polymer concentrations. This is because the amphiphilic PCLA copolymers alone or in the bioconjugates formed flower-like micelles when the concentration in the medium increased over the critical micelle concentration. The poly(caprolactone-*co*-lactide) block is hydrophobic, while the PEG block is hydrophilic in the copolymer that induces micelle formation. It is noteworthy that PEG also exhibited mild hydrophobic interactions with increasing temperature driven by the loss of orientational entropy in which the ordered interaction between PEG and water molecules was disordered. This not only resulted in entropy change (DS > 0), but also led to the hydrophobic aggregation. Thus, with the increase of the temperature, the hydrophobic aggregation was strengthened and under certain conditions such as concentrations, hydrophobic and hydrophilic balance, and medium PCLA and other poly(ester)-based copolymer underwent sol-to-gel phase transition (Figure 2). The sol-to-gel phase transition temperature of the PCLA copolymer and Lys-PCLA bioconjugate from the phase diagram was found to be 16.25 wt.% and 17.25 wt.%, respectively. Meanwhile, both materials exhibited good sol-to-gel phase transition and the gel window covered the body temperature, implying that the prepared thermo-responsive hydrogels are suitable for in vivo in situ gelation. At the low temperatures, the thermo-responsive sols could be mixed with therapeutic agents and injected into the body to form gel depot for the sustained release of therapeutic agents.

The representative sol-to-gel phase transition photograph of Lys-PCLA bioconjugate hydrogels prepared at 22.5 wt.% concentration is shown in Figure 3B. The PCLA copolymers induced the self-assembly in water at a low temperature and turned into immovable gel depot due to aggregation and the formation of clustered micelles at the physiological condition. More importantly, the formed hydrogel depots retained their stability after incubation with PBS over a period of 24 h, indicating the stability of Lys-PCLA hydrogels (Figure 3C). Therefore, Lys-PCLA hydrogel can be a suitable candidate for in vivo hydrogel depot for drug delivery and other biomedical applications. 

Hydrophilic properties of hydrogels could influence the behavior of hydrogels in biomedical applications. Therefore, the hydrophilic properties of hydrogels were investigated by measuring the water contact angle. The water contact angle of PCLA and Lys-PCLA hydrogels is shown in Figure 4. As expected, conjugation of Lys in the hydrogels slightly increased the hydrophilic characteristics of the gels.

### 3.3. Rheological Properties of Hydrogels

Rheological properties of hydrogels play an important role in in vivo and in vitro injectability. In order to ascertain viscosity and critical gelation temperature (T_gel_) of PCLA and Lys-PCLA hydrogels, the complex viscosity and modulus (G′ and G″) of hydrogels at different temperatures have been examined. As expected, at 22.5 wt.%, both PCLA and Lys-PCLA bioconjugates changed in viscosity with the increase of temperature (Figure 5A). At a low temperature, the viscosity of both samples showed a linear trend until 20 °C. Thereafter, the viscosity steadily increased and reached a maximum above 30 °C, indicating that Lys-PCLA bioconjugate formulations could be injected into the warm-blooded animal models for in situ hydrogel depot formation. When the temperature increased to about 40 °C, the viscosity of the PCLA hydrogels started decreasing due to the expulsion of the water from the hydrogel network and ultimately led to the contraction of the gels. Interestingly, the viscosity values attained a plateau for Lys-PCLA hydrogels, indicating that the Lys in the hydrogel network firmly holds the networks and maintains the integrity of the networks. This is mainly due to the stability of Lys protein at a higher temperature.

The T_gel_ was determined by the point at which temperature of the storage modulus (G′) and loss modulus (G″) crossover occurred. Figure 5B,C displays the typical elastic behavior at a higher temperature, verifying that G′ was always higher than G″. Such a behavior indicates the dominance of the elastic nature rather than the viscous nature of materials. As shown in the graph, the T_gel_ of Lys-PCLA and PCLA hydrogels was found to be 20.50 °C and 24.10 °C, respectively. The slightly low temperature gelation from Lys-PCLA hydrogels was due to non-covalent cross-linking between Lys and PCLA copolymers, which accelerated the gelation process. The mechanical properties of hydrogels are the key characteristics in tissue engineering and controlled delivery applications. In particular, the G′ of both Lys-PCLA and PCLA hydrogels was higher than the G″ when the temperature was raised over 20 °C, which implies that elastic modulus was higher at this temperature. The G′ value of Lys-PCLA at 37 °C was close to 10,000 Pa which showed the good stiffness of the gel similar to other reported hydrogel formulations [21].

Furthermore, oscillatory strain-sweep and frequency-sweep measurements were carried out as a function of shear strain and frequency, so as to determine the LVER and robustness of the hydrogels. The oscillatory strain-sweep test was performed on a preformed hydrogel at a constant frequency of 1 Hz at 37 °C from 1% to 100% strain amplitude. As shown in Figure 6A,B, both Lys-PCLA and PCLA hydrogels exhibit higher G′ values than G″ at lower shear strain, which further signifies that the hydrogels are predominantly elastic rather than viscous. However, both G′ and G″ decreased after certain shear strain, indicating the gel-sol transition. Interestingly, Lys-PCLA hydrogels could withstand a higher strain than PCLA hydrogels, which indicates the enhancement of their mechanical properties after conjugation with Lys. The frequency sweep test of the hydrogels was conducted at a fixed 4% strain, which is good enough to maintain the 3D structure of the hydrogel network. As shown in Figure 6C,D, the G′ of hydrogels was dependent on frequency, it slowly increased at a lower frequency, and elevated at a higher frequency, implying the presence of non-covalent interactions. Notably, the G′ of Lys-PCLA hydrogels was higher than PCLA hydrogels. This is mainly because the presence of Lys in the network effectively solvates and makes a inter- and intra-molecular hydrogen bonding network.

Hydrogels are generally considered as soft materials with viscoelastic behavior. The viscosity of both PCLA and Lys-PCLA hydrogels is low and shows no significant changes in viscosity between 0 °C and 20 °C, which indicates the Newtonian flowing of hydrogels. Interestingly, when the temperature was raised over 20 °C, the viscosity started increasing gradually and reached a maximum close to the physiological temperature. This suggested the non-Newtonian flowing of the hydrogels. Furthermore, the oscillatory temperature-sweep, strain-sweep and frequency-sweep tests are often employed to investigate the viscoelastic property of a hydrogel. In Figure 6, both G′ and G″ exhibit dependent temperature, with the appearance of a transition point from a viscous liquid-like state to a gel-like solid when G′ is crossed over G″. In addition, G′ and G″ also show strain and frequency dependence. Summarily, these gel-like materials reveal characteristics of a non-Newtonian liquid. 

### 3.4. In Vitro Biocompatibility of Hydrogels

Non-toxicity of a hydrogels is an essential factor for biomaterials in in vivo applications. To examine biocompatibility, different concentrations of PCLA copolymers or Lys-PCLA bioconjugates were exposed to RAW263.7 macrophages and their viability was evaluated with an MTT assay. Figure 7 shows the cell viability of RAW263.7 macrophages after co-culture with hydrogels. Both samples showed a high viability (>80%) even at 2000 µg/mL concentration, indicating the biocompatibility of the hydrogels.

In general, hydrogels exhibit numerous interesting properties such as non-toxicity, self-healing, biocompatibility, and biodegradability. Among them, biocompatibility is an important prerequisite for any hydrogels injected into the living body. Therefore, these characteristics are frequently demanded and the hydrogels with such properties are given attention in medical applications. When the hydrogels are applied into the body, the biocompatibility possible toxic effect of degraded byproducts need to be considered. In situ forming injectable hydrogels developed in this study are based on biodegradable poly(ester), such as PCLA. The PCLA-based hydrogels developed in this study show little or no toxicity at high concentrations (1000 µg/mL). This result was in agreement with our previous report based on PCLA hydrogels [57]. It should be noted that PCLA-based hydrogels implanted into the back of SD rats exhibited controlled degradation and no inflammation was found at the implantation site. In addition, the Lys used in this study is a naturally occurring protein found in the living organisms. Tan et al. developed PEG-lysozyme and this showed good biocompatibility and effectively sealed blood leakage in a rabbit trachea [44].

### 3.5. In Vivo In Situ Gelation

With the promising phase transition and good biocompatibility in vitro, the in situ gel formation was examined in warm-blooded animals. The in situ gel formation was investigated by locally injecting the Lys-PCLA bioconjugate sols or PCLA copolymer sols (300 µL, 22.5 wt.%) into the subcutaneous layer of SD rats using a 26 G hypodermic needle. The rats were sacrificed 10 min after injection and the injection site was cut open to investigate the gelation. As observed in Figure 8A,B, the free-flowing sols were effectively transformed into a hydrogel depot in the subcutaneous layers and showed effective adhesion and integration with the tissues. More importantly, the peeled gel from the tissues presented good stability and retained the structures for a few hours. Redness or bleeding at the injection site was also not observed, because the rapid gelation covers the hole formed during injection.

The subcutaneously formed hydrogels were recovered and subjected to SEM imaging to confirm surface morphology, porosity, and diffusivity of the hydrogels. The SEM images of both Lys-PCLA and PCLA hydrogels are shown in Figure 8C,D. According to the SEM micrographs, hydrogels are porous and show a compact mesh-like network. Particularly, the Lys-PCLA hydrogel shows a relatively homogenous porous structure, as opposed to PCLA hydrogels in which the pores are irregular and relatively less porous than Lys-PCLA hydrogels. The distinct porous pattern of Lys-PCLA hydrogels can be explained by the presence of cationic protein that induces electrostatic repulsion and allows the formation of uniform pores. The classical porous structural properties allow the intrusion of immune cells as well as the transport of nutrients and metabolites.

### 3.6. In Vivo Recruitment of Host Cells by Hydrogels

As shown in Figure 1, the aim of this hydrogel is to develop an injectable depot to invite DCs and intruded cells into the microporous network which can be activated by introducing appropriate antigens [39,58]. The ability of immune cells to intrude into the microporous network was examined in vivo by subcutaneous administration of PCLA copolymer or Lys-PCLA bioconjugate sols to form a hydrogel depot. Three days after injection, the nodules were retrieved to count the number of recruited cells and categorize the type of recruited cells by FACS analysis. From the quantitative analysis results, it was found that huge numbers of cells were migrated to the porous network in both formulations (Figure 9A). Both hydrogels showed recruitment of over a hundred million cells, implying that porous hydrogel networks are a good scaffold for host cell recruitment. Notably, Lys-PCLA hydrogels recruited more cells than the PCLA hydrogel. This is because the presence of the hydrophilic protein in the Lys-PCLA hydrogels induced a larger pore and thus allowed room to recruit more immune cells. Antigen presenting cells tend to bridge the adaptive and innate response. Hence, we analyzed the presence of CD11c+ DCs in the porous hydrogel network. As shown in Figure 9B, Lys-PCLA hydrogels recruited over 10 million CD11c+ DCs which is significantly higher than that of PCLA hydrogels. According to the in vivo cell recruitment results, the in situ forming injectable hydrogel provides a suitable microenvironment for the infiltration of host immune cells. The thermo-responsive smart materials were utilized to spontaneously assemble into a microporous hydrogel depot, which not only bypassed the ex vivo synthesis of the scaffold but also minimized the constraints associated with the preformed hydrogels.

Numerous locally injectable hydrogels have been developed for tumor immunotherapy. In particular, gelatin-based hydrogels are widely used in various biomedical applications because their properties often mimic the natural properties of the extracellular matrix. Various gelatin-based cell adhesive and degradable hydrogels have been developed and were shown to have good cell attachment, proliferation, and survival properties. Koshy et al. developed injectable porous gelatin cryogels and they were found to be minor host responsive following subcutaneous injection [59]. Controlled release of the granulocyte-macrophage colony-stimulating factor from gelatin hydrogels demonstrated effective infiltration of immune cells. It should be noted that, the Lys-PCLA hydrogels developed in this study show effective infiltration of immune cells without any adjuvants.

## 4. Conclusions

This study developed an in situ forming injectable Lys-PCLA bioconjugate hydrogel for the in situ recruitment of DCs. The presence of natural Lys protein in the hydrogel network endows a good porous mesh-like network. The Lys-PCLA hydrogel is non-toxic to macrophages even at 1000 µg/mL concentrations. Lys-PCLA hydrogels exhibited good sol-to-gel phase transition and good mechanical properties. The hydrogel endows a good in situ gel forming ability into the back of rats. Subcutaneous administration of Lys-PCLA bioconjugate sols into the back of mice spontaneously forms a hydrogel network. Owing to the porous properties, various immune cells including DCs and macrophages migrated to the porous network. The cells are modulated by the introduction of appropriate antigens, which can control the antitumor immunity. As suggested by the results, Lys-PCLA conjugate is a powerful alternative to therapies that require host cell modulation or the in situ reprogramming of host cells.

## Data Availability

The data that support the findings of this study are available from the corresponding authors, upon reasonable request.

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
