# Peer review of "Injectable Hydrogel Based on Protein-Polyester Microporous Network as an Implantable Niche for Active Cell Recruitment"

_pharmaceutics, 2022, doi:10.3390/pharmaceutics14040709_

Round 1

Reviewer 1 Report

The manuscript entitled ''Thermo-Responsive Biocomposite Injectable Hydrogel Based on Protein-Polyester Microporous Network as an Implantable Niche for Active Cell Recruitment'' contains some interesting findings, and it may ultimately be suitable for publication. Considerable attention was made to prepare the review, which is interesting. However, some issues need to be clarified before further consideration. Thus, the reviewer recommends this work can be published after a major review.

The reviewer has the following comments  

  1. Concise the title of the manuscript.
  2. City, state, country, etc., details of chemical provider should be reported. For instance, Sigma Aldrich, (city, state, country).
  3. All the chemical entities should have the names under each structure in Figure 1.
  4. The yield should be reported for all the chemical compounds/polymers
  5. What are the following values in Table of Figure 2?

2170-1500-2170 and 1912-1500-1912

  1. Statistical analysis should be moved before the results section.
  2. FTIR conditions were reported but the spectra details were not reported, it should be resolved.
  3. Thermo-mechanical properties of hydrogels should be measured using DSC and TGA
  4. The water contact angle is vital for determining hydrogel hydrophilicity or hydrophobicity, an essential factor in drug delivery, cell proliferation, and adherence. Therefore, the water contact angles of hydrogels should be reported in the revised manuscript.
  5. SEM images of cross-sectioned hydrogels should be added to the revised manuscript.
  6. The mechanical properties of hydrogels are the key characteristics of hydrogels in tissue engineering applications. Specifically, cyclic tensile and storage and loss modulus (G'&G'') are the main properties to describe hydrogel performance in tissue engineering. The authors should be compared the cyclic tensile and (G'&G'') data of PCLA and Lys-PCLA with the following results

https://doi.org/10.3390/ph14040291

  1. The introduction section should be revised entirely so that the reader can clearly identify the scientific problems solved by this research. There are several biocompatible systems with a wide range of biomaterials, including hydrogels, nanoparticles, nanofilms, nanocomposites, etc., but why did the authors select only hydrogels? The authors should emphasize why the hydrogels are familiar, or favor compared to other systems using the following articles. Moreover, the information on biomaterials (Chitosan, Gelatin, Zein, and PCL) should be elaborated in the introduction with recent references. Thus, the following articles should be quoted in the introduction.

Chitosan (https://doi.org/10.1016/j.colsurfb.2021.111819),

Gelatin (https://doi.org/10.3390/ph14040291,https://doi.org/10.1016/j.jmbbm.2020.103696) PCL (https://doi.org/10.1016/j.msec.2020.110928), https://doi.org/10.3390/pharmaceutics11120621

It would be more realistic to cover such kind of research work in the current manuscript. Which will enrich the quality of the current manuscript as well as inquisitiveness to the readers.

  1. According to the revised data, the conclusions should be modified with more quantitative data.

Author Response

Reviewer #1

The manuscript entitled ''Thermo-Responsive Biocomposite Injectable Hydrogel Based on Protein-Polyester Microporous Network as an Implantable Niche for Active Cell Recruitment'' contains some interesting findings, and it may ultimately be suitable for publication. Considerable attention was made to prepare the review, which is interesting. However, some issues need to be clarified before further consideration. Thus, the reviewer recommends this work can be published after a major review. The reviewer has the following comments.

  1. Concise the title of the manuscript.

Response: As the reviewer has suggested, we given a concise title to our manuscript. The title “''Thermo-Responsive Biocomposite Injectable Hydrogel Based on Protein-Polyester Microporous Network as an Implantable Niche for Active Cell Recruitment” has been replaced with “Injectable Hydrogel Based on Protein-Polyester Microporous Network as an Implantable Niche for Active Cell Recruitment” in the revised manuscript.

  1. City, state, country, etc., details of chemical provider should be reported. For instance, Sigma Aldrich, (city, state, country).

Response: We thank the reviewer for noticing this error. We included the city, state, country, etc., details of chemicals and instrument provider in the revised manuscript. The details described in the section Materials and Methods page 3 and line 109.

  1. All the chemical entities should have the names under each structure in Figure 1.

Response: As the reviewer has suggested, names of each chemical structures were provided in the revised manuscript Figure 1 as follows:

  1. The yield should be reported for all the chemical compounds/polymers.

Response: As pointed out by the reviewer, the obtained yield of all the copolymers and conjugates has been provided in the revised manuscript (page 3 and 4).  

  1. What are the following values in Table of Figure 2? 2170-1500-2170 and 1912-1500-1912

Response: The Table of Figure 2 shows the molecular structure of PCLA-PEG-PCLA copolymers. The molecular structure of PCLA-PEG-PCLA copolymers were determined using 1H NMR spectra and GPC. To clarify this, we separated the Table from Figure 2 and presented with more details in page 8 line 268.

  1. Statistical analysis should be moved before the results section.

Response: As the reviewer has suggested, the statistical analysis information has been moved before the results section (page 6 and line 218).

  1. FT-IR conditions were reported but the spectra details were not reported, it should be resolved.

Response: We thank the reviewer for noticing this error. We included the FT-IR results of copolymers (newly inserted Figure 2d) and modified the Figure 2 in the revised manuscript follows:

  1. Thermo-mechanical properties of hydrogels should be measured using DSC and TGA.

Response: As pointed out by the reviewer, it will be interesting to observe the thermo-mechanical properties of hydrogels using DSC and TGA analysis. However, the injectable hydrogels prepared in this study used between room and body temperature. In this study as well as from our previous studies, we observed the good sol-to-gel phase transition between room and body temperature. More importantly, the implanted hydrogel depot in the warm-blooded animals exhibited controlled biodegradation and completely degraded within 6 weeks after implantation (Biomaterials 230 (2020) 119599, Carbohydrate Polymers 233 (2020) 115832, Journal of Industrial and Engineering Chemistry 96 (2021) 345–355). Therefore, we didn’t measure the DSC and TGA. It will be interesting to observe the thermo-mechanical properties of Lys-PCLA hydrogels using DSC and TGA analysis, which will be investigated as further study.

  1. The water contact angle is vital for determining hydrogel hydrophilicity or hydrophobicity, an essential factor in drug delivery, cell proliferation, and adherence. Therefore, the water contact angles of hydrogels should be reported in the revised manuscript.

Response: We agree with the reviewer’s comments that water contact angle is vital for determining hydrogel hydrophilicity or hydrophobicity, an essential factor in drug delivery, cell proliferation, and adherence. Therefore, water contact angle of hydrogels has been measured and included in the revised manuscript. In order to include information on contact angle details, we inserted sentences in the revised manuscript page 9 and line 311 as follows:

“Hydrophilic properties of hydrogels could influence the behavior hydrogels in biomedical applications. Therefore, hydrophilic properties of hydrogels were investigated by measuring the water contact angle. The water contact angle of PCLA and Lys-PCLA hydrogels was shown in Figure. As expected, conjugation of Lys in the hydrogels slightly increased the hydrophilic characteristics of gels.”

Figure 4. Water contact angle of PCLA and Lys-PCLA hydrogels. Data are presented as mean ± SD (n=4).

  1. SEM images of cross-sectioned hydrogels should be added to the revised manuscript.

Response: The SEM images in Figure 7C and 7D are for the cross-sectional image of the Lys-PCLA and PCLA hydrogels. To observe the porous structure of the hydrogels, the gels were recovered from the SD rats 10 min after implantation and frozen using liquid nitrogen before being free-dried. Thereafter, the gels the freeze-dried hydrogels were cross-sectioned and coated with platinum using spin coater and then the porous structure was visualized using SEM. In order to clearly describe this information, we modified few sentences in the revised manuscript Page 6 and line 204.

  1. The mechanical properties of hydrogels are the key characteristics of hydrogels in tissue engineering applications. Specifically, cyclic tensile and storage and loss modulus (G'&G'') are the main properties to describe hydrogel performance in tissue engineering. The authors should be compared the cyclic tensile and (G'&G'') data of PCLA and Lys-PCLA with the following results (https://doi.org/10.3390/ph14040291).

Response: We agree with the reviewer’s comments that mechanical properties of hydrogels are the key characteristics of hydrogels in tissue engineering applications. The storage and loss modulus (G'&G'') of hydrogels clearly describes the elastic (gel) and viscous (liquid) of behavior. As pointed out by the reviewer, the G'&G'' of PCLA-based hydrogels were compared with other biocompatible hydrogel system such as GelMA-based nanocomposite hydrogel systems with appropriate supporting references.

As the reviewer has suggested, we have inserted the mechanical properties comparison in the revised manuscript in page 10 line 342 as follows:

“The mechanical properties of hydrogels are the key characteristics in tissue engineering and controlled delivery applications. In particular, the G' of both Lys-PCLA and PCLA hydrogels was higher than G'' when the temperature was raised over 20 °C, which implied that elastic modulus was higher at this temperature. The G' value Lys-PCLA at 37 °C was close to 10,000 Pa that showed the good stiffness of the gel similar with other reported hydrogel formulations [21].”

  1. The introduction section should be revised entirely so that the reader can clearly identify the scientific problems solved by this research. There are several biocompatible systems with a wide range of biomaterials, including hydrogels, nanoparticles, nanofilms, nanocomposites, etc., but why did the authors select only hydrogels? The authors should emphasize why the hydrogels are familiar, or favor compared to other systems using the following articles. Moreover, the information on biomaterials (Chitosan, Gelatin, Zein, and PCL) should be elaborated in the introduction with recent references. Thus, the following articles should be quoted in the introduction. Chitosan (https://doi.org/10.1016/j.colsurfb.2021.111819), Gelatin (https://doi.org/10.3390/ph14040291, https://doi.org/10.1016/j.jmbbm.2020.103696) PCL (https://doi.org/10.1016/j.msec.2020.110928), https://doi.org/10.3390/pharmaceutics11120621

It would be more realistic to cover such kind of research work in the current manuscript. Which will enrich the quality of the current manuscript as well as inquisitiveness to the readers.

Response: We agree with the reviewer’s comments that there are several biocompatible systems, including hydrogels, nanoparticles, nanofilms, and nanocomposites, available to recruit immune cells and subsequently activate the immune cells. In this study, we chose in situ forming injectable hydrogel system as a representative biomaterial to recruit and activate immune cells due to their unique properties. In general, the in situ implanted hydrogels containing appropriate antigens or cytokines effectively recruit, migrate, and activate immune cells in human body. This is unlike ex vivo immune modulation of cancer patients that require complex designs, in which patients’ own immune cells are isolated and activated ex vivo. Through the use of injectable hydrogels, the free-flowing sols could be injected into the patient of interest that could generate antigen-specific immune response via the introduction of appropriate antigens. The release of antigens from the microporous hydrogel network can stimulate antigen-presenting cells, including dendritic cells (DCs), macrophages, and B cells.

As pointed out by the reviewer, hydrogels prepared using chitosan, gelatin, Zein and PCL have been extensively investigated for various biomedical application because of their biocompatibility, non-immunogenicity and stability at the physiological conditions.

As the reviewer has suggested, we have inserted the discussion of various hydrogels in the revised manuscript in page 2 line 55 as follows:

“Hydrogels prepared using natural polymers such as chitosan, zein and gelatin by cross-linking exhibited good stimuli-responsive characteristics and controlled release characteristics [20-23]. Poly(caprolactone)-based synthetic hydrogels showed sharp sol-to-gel phase transition and prolonged the release of therapeutics [24].”

  1. According to the revised data, the conclusions should be modified with more quantitative data.

Response: As the reviewer has suggested, the conclusion has been modified with more quantitative data.

We appreciate reviewers’ valuable comments and hope that the revised manuscript is suitable for publication in Pharmaceutics.

Reviewer 2 Report

This manuscript presents a work of preparing an implantable hydrogel based on protein-polyester as an implantable niche for cells recruitment. The manuscript is well written and the experiments are very well conducted.

I recommend acceptance of the article after performing the following:

1- Please discuss the biocomptability of the developed niche in more details.

2- Please determine the type of flow of the hydrogel accoriding to the viscosity plots obtained.

3- The PCLA cell viability values showed a significant decrease (below 80%) starting from 250 ug/ml concentration. Please discuss.

4- Please shed more lights on the obtained benefits of using such new hydrogels as compared to gelatin-based couterparts for example.

Author Response

Reviewer #2

This manuscript presents a work of preparing an implantable hydrogel based on protein-polyester as an implantable niche for cells recruitment. The manuscript is well written and the experiments are very well conducted.

I recommend acceptance of the article after performing the following:

  1. Please discuss the biocompatibility of the developed niche in more details.

Response: In general, hydrogels exhibit numerous interesting properties such as non-toxicity, self-healing, biocompatibility, and biodegradability. Among them, biocompatibility is an important prerequisite for any hydrogels injected into the living body. Therefore, these characteristics are frequently demanded and the hydrogels with such properties are gained attention in medical applications. When the hydrogels are applied into the body, in addition to the biocompatibility possible toxic effect of degraded byproducts need to be considered. In situ forming injectable hydrogels developed in this study are based biodegradable poly(ester), such as PCLA. The PCLA based hydrogels developed in this study shows no or little toxicity at high concentration (1,000 µg/mL). This result was in agreement with our previous report based on PCLA hydrogels (Biomaterials 230 (2020) 119599, Carbohydrate Polymers 233 (2020) 115832, Journal of Industrial and Engineering Chemistry 96 (2021) 345–355). It should be noted that PCLA based hydrogels implanted into the back of SD rats exhibited controlled degradation and no inflammation was found at the implantation site. In addition, the Lys used in this study is a naturally occurring protein found in the living organisms. Tan et al. developed PEG-lysozyme showed good biocompatibility and effectively sealed blood leakage in a rabbit trachea.

As the reviewer has suggested, we have inserted the discussion of biocompatibility hydrogels in the revised manuscript in page 12 line 391 as follows:

“In general, hydrogels exhibit numerous interesting properties such as non-toxicity, self-healing, biocompatibility, and biodegradability. Among them, biocompatibility is an important prerequisite for any hydrogels injected into the living body. Therefore, these characteristics are frequently demanded and the hydrogels with such properties are gained attention in medical applications. When the hydrogels are applied into the body, in addition to the biocompatibility possible toxic effect of degraded byproducts need to be considered. In situ forming injectable hydrogels developed in this study are based biodegradable poly(ester), such as PCLA. The PCLA based hydrogels developed in this study shows no or little toxicity at high concentration (1,000 µg/mL). This result was in agreement with our previous report based on PCLA hydrogels [57]. It should be noted that PCLA based hydrogels implanted into the back of SD rats exhibited controlled degradation and no inflammation was found at the implantation site. In addition, the Lys used in this study is a naturally occurring protein found in the living organisms. Tan et al. developed PEG-lysozyme showed good biocompatibility and effectively sealed blood leakage in a rabbit trachea [44].”

  1. Please determine the type of flow of the hydrogel according to the viscosity plots obtained.

Response: Hydrogels are generally considered as soft materials with viscoelastic behavior. The viscosity of both PCLA and Lys-PCLA hydrogels is low and shows no significant changes in viscosity between 0 °C to 20 °C, which indicated Newtonian flowing of hydrogels. Interestingly, when the temperature raised over 20 °C, viscosity started increasing gradually and reached maximum close to the physiological temperature. This suggested the non-Newtonian flowing of the hydrogels. Furthermore, the oscillatory temperature-sweep, strain-sweep and frequency-sweep tests are often employed to investigate the viscoelastic property of hydrogel. In Figure 4, both G’ and G’’ exhibit dependent temperature, with the appearance of a transition point from viscous liquid-like state to gel-like solid when G’ is crossed over G’’. In addition, G’ and G’’ also show the strain and frequency dependence. Summarily, these gel-like materials reveal characteristics of a non-Newtonian liquid.

As pointed out by the editor, to increase the scientific novelty of the manuscript, we discussed the flow of the hydrogel in the revised manuscript in page 12 line 372 as follows:

“Hydrogels are generally considered as soft materials with viscoelastic behavior. The viscosity of both PCLA and Lys-PCLA hydrogels is low and shows no significant changes in viscosity between 0 °C to 20 °C, which indicated Newtonian flowing of hydrogels. Interestingly, when the temperature raised over 20 °C, viscosity started increasing gradually and reached maximum close to the physiological temperature. This suggested the non-Newtonian flowing of the hydrogels. Furthermore, the oscillatory temperature-sweep, strain-sweep and frequency-sweep tests are often employed to investigate the viscoelastic property of hydrogel. In Figure 4, both G’ and G’’ exhibit dependent temperature, with the appearance of a transition point from viscous liquid-like state to gel-like solid when G’ is crossed over G’’. In addition, G’ and G’’ also show the strain and frequency dependence. Summarily, these gel-like materials reveal characteristics of a non-Newtonian liquid.”

  1. The PCLA cell viability values showed a significant decrease (below 80%) starting from 250 ug/ml concentration. Please discuss.

Response: As pointed out by the reviewer, the cell viability of PCLA copolymers are decreased when they were co-incubated with over 250 µg/mL of copolymers. Precisely, the cell viability of PCLA copolymer at 500 µg/mL and 1,000 µg/mL concentration was found to be 81.2±2 and 78.2±3, respectively. At high PCLA copolymer concentrations, the cells viability started decreased probably due to the slightly more hydrophobic balance of copolymers. After certain increase in PCLA copolymer concentrations (over 500 µg/mL), the cell viability not significantly decreased, indicating the superb biocompatibility of PCLA copolymers. It should be noted that the cell viability of PCLA copolymers was significantly higher than of other reported poly(ester)-based copolymers.

  1. Please shed more lights on the obtained benefits of using such new hydrogels as compared to gelatin-based counterparts for example.

Response: Gelatin-based hydrogels are widely used in various biomedical applications because their properties often mimic the natural properties of extracellular matrix. Tunable physical and biological properties of gelatin-based hydrogels, and the presence of cell-adhesive peptide motifs allow the cells proliferate and spread in the gelatin-based scaffolds. Various gelatin-based cell adhesive and degradable hydrogels have been developed and were shown good cell attachment, proliferation, and survival properties. Koshy et al. developed injectable porous gelatin cryogels and found to elicit minor host responsive following subcutaneous injection. Controlled release of granulocyte-macrophage colony-stimulating factor from gelatin hydrogels shown effective infiltration of immune cells. It should be noted that, the Lys-PCLA hydrogels developed in this study shows effective infiltration of immune cells without any adjuvants.

As pointed out by the editor, to increase the scientific novelty of the manuscript, we have compared the gelatin-based hydrogels with the newly developed hydrogels in the revised manuscript in page 14 line 470 as follows:

“Numerous locally injectable hydrogels have been developed for tumor immunotherapy. In particular, gelatin-based hydrogels are widely used in various biomedical applications because their properties often mimic the natural properties of extracellular matrix. Various gelatin-based cell adhesive and degradable hydrogels have been developed and were shown good cell attachment, proliferation, and survival properties. Koshy et al. developed injectable porous gelatin cryogels and found to elicit minor host responsive following subcutaneous injection. Controlled release of granulocyte-macrophage colony-stimulating factor from gelatin hydrogels shown effective infiltration of immune cells. It should be noted that, the Lys-PCLA hydrogels developed in this study shows effective infiltration of immune cells without any adjuvants.”

We appreciate reviewers’ valuable comments and hope that the revised manuscript is suitable for publication in Pharmaceutics.

Round 2

Reviewer 1 Report

The authors have clarified all my concerns, and the quality of the manuscript was ameliorated. I must congratulate the authors for their willingness in addressing the reviewer’s comments. So, I recommend accepting the manuscript in its present form.